# Synthesis and Validation of a Bioinspired Catechol-Functionalized Pt(IV) Prodrug for Preclinical Intranasal Glioblastoma Treatment

**DOI:** 10.3390/cancers14020410

**Published:** 2022-01-14

**Authors:** Xiaoman Mao, Shuang Wu, Pilar Calero-Pérez, Ana P. Candiota, Paula Alfonso, Jordi Bruna, Victor J. Yuste, Julia Lorenzo, Fernando Novio, Daniel Ruiz-Molina

**Affiliations:** 1Catalan Institute of Nanoscience and Nanotechnology (ICN2), CSIC and BIST, Campus UAB, Bellaterra, 08193 Barcelona, Spain; mmchong7@zju.edu.cn (X.M.); paula.alfonso@icn2.cat (P.A.); 2Departament de Bioquímica i Biologia Molecular, Universitat Autònoma de Barcelona, Cerdanyola del Vallès, 08193 Barcelona, Spain; psychews@gmail.com (S.W.); pilar.calero@uab.cat (P.C.-P.); AnaPaula.Candiota@uab.cat (A.P.C.); julia.lorenzo@uab.cat (J.L.); 3Centro de Investigación Biomédica en Red: Bioingeniería, Biomateriales y Nanomedicina, Cerdanyola del Vallès, 08193 Barcelona, Spain; 4Institut de Biotecnologia i Biomedicina, Universitat Autònoma de Barcelona, Cerdanyola del Vallès, 08193 Barcelona, Spain; 5Neuro-Oncology Unit, Bellvitge University Hospital-ICO (IDIBELL), Avinguda de la Gran Via de l’Hospitalet, 199-203, L’Hospitalet de Llobregat, 089098 Barcelona, Spain; jbruna@bellvitgehospital.cat; 6Institut de Neurociències, Departament de Bioquimica i Biologia Molecular, Facultat de Medicina, Universitat Autònoma de Barcelona (UAB), Campus UAB, Cerdanyola del Vallès, 08193 Barcelona, Spain; Victor.Yuste@uab.cat; 7Departament de Química, Universitat Autònoma de Barcelona (UAB), Campus UAB, Cerdanyola del Vallès, 08193 Barcelona, Spain

**Keywords:** glioblastoma, bioinspired, Pt(IV), prodrug, catechol, platinum drugs, intranasal

## Abstract

**Simple Summary:**

Glioblastoma (GB) is a type of brain cancer with a poor prognosis and few improvements in its treatment. One of the greatest difficulties in GB therapy lies in the fact that most of the drugs with high anticancer potential do not reach the brain and exert high therapeutic activity while minimizing side effects. To overcome these limitations, we focused on a catechol-based Pt(IV) prodrug (able to reverse cisplatin in a cellular environment) with the intention of repurposing Pt-based drugs as GB chemotherapeutic agents. Our in vitro results have corroborated the therapeutic effect of the synthesized complexes as comparable to cisplatin, and in vivo studies have demonstrated the potential of nose-to-brain delivery of this Pt(IV) prodrug for GB treatment.

**Abstract:**

Glioblastoma is the most malignant and frequently occurring type of brain tumors in adults. Its treatment has been greatly hampered by the difficulty to achieve effective therapeutic concentration in the tumor sites due to its location and the blood–brain barrier. Intranasal administration has emerged as an alternative for drug delivery into the brain though mucopenetration, and rapid mucociliary clearance still remains an issue to be solved before its implementation. To address these issues, based on the intriguing properties of proteins secreted by mussels, polyphenol and catechol functionalization has already been used to promote mucopenetration, intranasal delivery and transport across the blood–brain barrier. Thus, herein we report the synthesis and study of complex **1**, a Pt(IV) prodrug functionalized with catecholic moieties. This complex considerably augmented solubility in contrast to cisplatin and showed a comparable cytotoxic effect on cisplatin in HeLa, 1Br3G and GL261 cells. Furthermore, preclinical in vivo therapy using the intranasal administration route suggested that it can reach the brain and inhibit the growth of orthotopic GL261 glioblastoma. These results open new opportunities for catechol-bearing anticancer prodrugs in the treatment for brain tumors via intranasal administration.

## 1. Introduction

Glioblastoma (GB) is the most frequent malignant primary brain tumor in adults with the poorest prognosis and a median overall survival below 2 years, even after treatments combining gross macroscopic resections, radiotherapy and chemotherapy [1,2]. This, together with its aggressiveness and almost ensured recurrence, has driven many efforts over recent years to advance its treatment, unfortunately without germane results. The main challenge for GB drugs is to cross the blood–brain barrier (BBB), together with a restricted bioavailability that prevents achieving the right dose, a common limitation for treating neurological CNS disorders [3]. Temozolomide (TMZ) merely showed the ability to cross the BBB and reach the tumor with the required therapeutic levels as demonstrated in a clinical trial back in 2005, especially in the subgroup of MGMT gene promoter methylated GBs [4,5,6]. Since then, trials with different systemic treatments at maximum tolerated doses have been unsuccessfully tested, leading to a sequence of side effects [7,8], as well as to the rapidly gained resistance of GB, in part related to the requested rest treatment intervals [9]. So, achieving suitable therapeutic doses within brain tumors while avoiding severe secondary effects turned into a challenge, even for aggressive gliomas that compromise BBB integrity [10].

One of the emerging strategies is to overcome this limitation via the non-invasive and easily accessible intranasal (IN) administration [11]. Over the last decade, clinical trials including patients of all group ages, genders and clinical phases experienced a threefold increase compared with the previous 2000–2010 decade (see ClinicalTrial.gov, accessed on 24 November 2021) [12]. In addition to bypassing the BBB, IN-administered drugs with systemic action can be rapidly absorbed through rich vasculature in the submucosa and exhibit a fast onset of action [13], positioning this delivery route into a theoretical privileged position to fight GB [14]. However, its implementation in commercial uses still remains elusive, among others, due to a reduced mucopenetration and rapid mucociliary clearance of most effective drugs tested. Therefore, syntheses of novel therapeutic agents aimed to overcome these limitations are intensely sought.

Herein, we hypothesize that this objective can be achieved with the synthesis of GB-active drugs bearing catechol groups, which are expected to enhance the drug mucopenetration bioinspired by the mussel adhesion properties in seawater environments [15]. Following this hint, films for effective buccal drug delivery [16], chitosan-catechol polymers with long-lasting mucoadhesive properties [17] or polydopamine (PDA) coatings with enhanced mucopenetration and cellular uptake [17,18] have already been reported. This is attributed to their hydrophilic characteristics and negative charge at physiological pH that minimizes the interaction with the negatively charged and hydrophobic pockets in mucus, a prerequisite for an efficient mucopenetration [19,20,21,22].

As a proof of concept to validate our approach, we have designed complex **1**. While effective for most malignant tumors [23], platinum-based drugs are only used as a residual fourth-line chemotherapeutics for GB, i.e., only when all standard therapeutic protocols fail. Thus far, reported platinum clinical trials have not improved the overall survival of patients with brain tumors [24], neither in combination with radiotherapy [25] nor other chemotherapeutics such as carmustine (BCNU) [26] and TMZ [27]. Therefore, if successful, our work would not only demonstrate the legitimacy of IN administration but also repurpose this family of drugs back to the first line of action. Pt(IV) agents also minimize side effects as they likely become active upon intracellular reduction to the active Pt(II) form, which interacts with DNA [28] and perturbs other pathways related to cisplatin resistance [29] by biological reductants [30], such as reduced nicotine adenine dinucleotide (NADH) [31], ascorbic acid [28] and even reduced glutathione (GSH) [32]. On top of that complex **1** mimics natural products such as bis-catechol gossypol or nordihydroguaiaretic acid (NDGA), which are already used as antioxidants and anti-inflammatories in cell culture and preclinical GB studies [33,34]. In fact, the development of novel bis-catechol with improved and reproducible therapeutic GB efficiencies has become a research line of interest per se; such is the case of terameprocol, which is considered a global transcription inhibitor that affects cell division, apoptosis, drug resistance and radiation resistance in hypoxia [35].

A schematic representation of the approach used for the validation of complex **1** as an efficient IN drug for the treatment of GB is shown in Figure 1. It mainly involves the synthesis of complex **1** and the in vitro testing with murine GB cells, using cisplatin (CDDP), NDGA and the precursor complex disuccinatocisplatin (DSCP) for comparison purposes. Finally, the in vivo tolerability test and therapeutic activity of complex **1** after intranasal administration was positively evaluated to confirm its efficacy.

## 2. Materials and Methods

### 2.1. Chemicals and Instrumentation

Unless otherwise noted, materials obtained from suppliers were used as received. ^1^H, ^13^C and ^1^H-^1^H COSY NMR spectra were recorded at 400, 100, and 400 MHz, respectively, using Bruker DPX 400 MHz spectrometer. ^195^Pt NMR spectrum was performed in a Bruker Avance DRX-360 spectrometer operating at 77.42 MHz. ^195^Pt chemical shifts are relative to external 1 M Na_2_[PtCl_6_] in D_2_O (0 ppm). High-resolution mass spectra were obtained by direct injection of the sample with electrospray techniques in a Bruker microTOF-Q instrument. FT-IR spectra were recorded using a Tensor 27 spectrophotometer (Bruker Optik GmbH, Leipzig, Germany) with KBr pellets. Sixty-four scans were collected for each sample at a resolution of 4 cm^−1^ over 400–4000 cm^−1^. PrestoBlue assays were carried out using a Victor 3 microplate reader (PerkinElmer, Boston, MA, USA), Statistical analysis and IC_50_ calculations were performed with GraphPad Prism (version 7.0, GraphPad Software Inc., San Diego, CA, USA). DNA concentration was determined by a NanoDrop UV-vis spectrometer (Thermo Scientific, California, CA, USA). ICP-MS (PerkinElmer, Waltham, MA, USA) was used to determine the metallic concentration.

### 2.2. Synthesis of Complex **1**

The synthesis of the intermediate DSCP was duplicated according to previously reported procedures [36]. DSCP (0.5 g, 0.93 mmol), *N,N*′-Dicyclohexylcarbodiimide (DCC, 0.58 g, 2.8 mmol) and N-Hydroxysuccinimide (NHS, 0.43 g, 3.75 mmol) were dissolved in anhydrous dimethylformamide (DMF, 4 mL) under Ar in darkness. After the solution was stirred overnight, the slurry mixture formed was cooled to 0 °C, and the white precipitate dicyclohexylurea (DCU) was removed by filtration. A large amount of ethyl acetate (EtOAc) was poured into the filtrate and sonicated to achieve an off-white precipitate, which was washed with cold ether and dried to give the activated ester (0.424 g, 62.2%). This intermediate (0.42 g, 0.58 mmol) and dopamine hydrochloride (0.263 g, 1.39 mmol) were dissolved in anhydrous DMF (3 mL), and N-methylmorpholine (NMM, 2.3 mL, 20.9 mmol) was added slowly. The reaction was tracked by thin-layer chromatography (TLC). After 24 h, the reaction was stopped and lyophilized to give a brownish residue, which was purified by column chromatography (EtOAC/MeOH, 9/1, *v*/*v*) to provide complex **1** as an off-white solid (0.195 g, 41.8%). ^1^H NMR (400 MHz, MeOD-d_4_): δ = 6.69 (m, 4H), 6.54 (d, 2H), 3.36 (t, 2H) 2.61–2.70 (m, 8H), 2.46 (t, 4H) ppm; ^13^C NMR (100 MHz, MeOD-d_4_): δ 180.9, 173.8, 144.7, 143.2, 130.79, 119.7, 115.5, 114.9, 41.1, 34.4, 31.3, 31.0; ^195^Pt NMR, δ (ppm) = 1012. HR-ESI-MS (*m*/*z*): [M + Na]^+^ calculated 827.1276, found 827.1182; Pt content by ICP-MS: calculated (%) 24.25, found (%) 24.12. Elem. Anal. Calcd. For C_24_H_34_O_10_N_4_Cl_2_Pt: C, 35.83; H, 4.26; N, 6.96, found: C, 35.51; H, 4.50; N, 6.51.

### 2.3. In Vitro Studies

#### 2.3.1. Cell Viability

Human cell lines (HeLa, 1Br3G and GL261) were purchased from the American Type Culture Collection (ATCC) and cultured at 37 °C in a humidified 5% CO2 environment in RPMI 1640 (GL261) supplemented with 2 mM of L-glutamine, 2 g/L of D-glucose and 10% fetal bovine serum (FBS); or Dulbecco Modified Eagle’s Medium (DMEM) supplemented with 2 mM of L-glutamine, 4 g/L of D-glucose and 10% FBS. Cells in exponential growth were seeded on 96-well plates (Corning, Tewksbury, MA, USA) at a density of 4000 cell/well. After 24 h incubation, fresh media containing compounds (CDDP, complex **1**, DSCP and NDGA) at different concentrations were added, and the plates were incubated for 24 or 72 h. Afterwards, 10 μl of PrestoBlue^®^ (0.15 mg/mL, Thermo Scientific, California, CA, USA) were added into each well, and the plates were incubated for another 4 h before measuring the fluorescence at 572 nm with excitation at 531 nm in a Victor 3 microplate reader (PerkinElmer, Germany). All experiments were carried out in triplicate, and the IC_50_ value for each compound was determined using GraphPad Prism software (version 7.0, GraphPad Software Inc., San Diego, CA, USA).

#### 2.3.2. Estimation of ROS Formation

HeLa, 1Br3G and GL261 cells were seeded on black 96-well microplates with optically clear bottoms (Corning, USA) at a density of 20,000 cells/well overnight. Cells were then washed, and PBS was added with the fluorescent probe DCFCDA at a final concentration of 10 μM for 30 min in the dark. Afterwards, the buffer with a probe was replaced by media or media with compounds (H_2_O_2_, complex **1** and CDDP) for 24 h. A total of 0.1 mM of H_2_O_2_ was used as a positive control, while other agents were added at their IC50 concentration. The fluorescence increase in each well was estimated by a microplate reader Victor 3 at 530 nm (emission) and 485 nm (excitation) (Perkin Elmer, USA). This experiment was repeated in triplicate independently.

#### 2.3.3. Cellular Internalization

GL261 cells were seeded in 6-well plates (Corning, USA) at a density of 300,000 cells per well. After 24 h incubation, the media were replaced by fresh media in absence or presence of CDDP and complex **1** at a final concentration of 10 μM Pt. The cells were allowed to internalize the compounds for different time points, then rinsed with PBS. Then, cells were harvested and digested, and the platinum content determined by ICP-MS.

#### 2.3.4. DNA-Bound Pt

To investigate the Pt content associated with the DNA of GL261 cells, similarly to the procedure for cell uptake, cells were harvested after incubation with CDDP and complex **1** at a concentration of 10 μM Pt for 24 h. The cellular pellets were resuspended and incubated on ice for 15 min in lysis buffer (150 mM of Tris-HCl (pH 8.0), 100 mM of NaCl and 0.5% (*w*/*v*) SDS). After centrifuge, RNase A was added at 200 μg/mL and incubated for 1 h at 37 °C. Later, proteinase K was added at 100 μg/mL to incubate for 3 h at 56 °C. Afterwards, a volume of phenol–chloroform (1/1, *v*/*v*) was added to the extract. Obtained aqueous phase containing DNA was collected and precipitated with 0.1 volume of 3 M of sodium acetate and 1 volume of absolute ethanol at −20 °C overnight. DNA samples were washed with ethanol, dried and resuspended in TE buffer (10 mM of Tris-HCl, 1 mM of EDTA, pH 8.0). DNA concentration of the sample was determined by spectral photometry (NanoDrop 1000, Thermo Scientific, USA). Quantified DNA was then frozen for analysis by ICP-MS.

### 2.4. In Vivo Studies

#### 2.4.1. Animal Studies

All animal experiments were approved by the local ethics committee (Comissió d’Ètica en l’Experimentació Animal i Humana, https://www.uab.cat/etica-recerca/, accessed on 24 November 2021), and conducted under the principles of regional and state legislations (protocol CEEAH 4859). Healthy female C57BL/6J mice (8–12 weeks, body weight 20–24 g) were used for all in vivo studies in this work and were obtained from Charles River Laboratories (Charles River Laboratories International, Saint-Germain-sur-l’Arbresle, France). Mice were housed at Servei d’Estabulari (animal facility) from Universitat Autònoma de Barcelona (UAB), and their health status was supervised by qualified staff. Each mouse was identified by a unique ear notch combination in order to distinguish animals in the same cage. Animals are then attributed an individual numeric code (e.g., tumor-bearing animals have the code CXXXX where XXXX is a sequential number) allowing not only for animal identification but also to know the implantation round used for its generation.

#### 2.4.2. Tolerability Assessment

The mice were randomly divided into two groups, control and complex **1**, *n* = 3 each. Anesthetized mice were intranasally administered with PBS or complex **1** using a micropipette (Gilson, Limburg, Germany) in a horizontally supine position. In total 3 doses were given, which increased from 0.9, 1.2 and 1.5 mg Pt/kg body weight every week [37]. The body weights of mice were recorded prior to the first dose of treatment and were monitored 3 times a week thereafter. The animals were closely observed by veterinary staff from the animal facility, in particular food/water consumption, behavior changes and possible signs of suffering [38]. The whole study lasted for 4 weeks. Mice were euthanized for humanitarian reasons when endpoint was reached.

#### 2.4.3. Orthotopic Intracranial GB Model by Stereotactic Injection of GL261 Cells

The GL261 GB model was generated by stereotactic injection of GL261 cells into C57BL/6J mice, performed by accredited personnel. Analgesia (Metacam, Ingelheim am Rhein, Germany) at 1 mg/kg was injected subcutaneously to each animal 15 min before anesthesia and also 24 and 48 h after implantation. Mice were anesthetized with a mixture of ketamine (Parke-Davis SL, Madrid, Spain) at 80 mg/kg and xylazine (Carlier, Barcelona, Spain) at 10 mg/kg via intraperitoneal administration. Once anesthetized, the mouse was immobilized on the stereotaxic holder (Kopf Instruments, Tujunga, CA, USA) in a prone position. Next, the head area was shaved, and the incision site was sterilized with iodophor disinfectant solution, a 1 cm incision was made exposing the skull, and a 1 mm hole was drilled 0.1 mm posterior to the bregma and 2.32 mm to the right of the midline using a microdrill (Fine Science Tools, Heidelberg, Germany). A 26G Hamilton syringe (Reno, NV, USA), positioned on a digital push–pull microinjector (Harvard Apparatus, Holliston, MA, USA) was then used for injection of 4 μL of RPMI cell culture medium containing 100,000 GL261 cells at a depth of 3.35 mm from the skull surface at a 2 μL/min.

Once the injection was completed, the Hamilton syringe was left untouched for 2 min more before its removal to prevent the cellular liquid leakage out of the skull. Finally, the Hamilton syringe was gently and slowly withdrawn, and the scission site was closed with suture silk 6.0 (Braun, Barcelona, Spain). When the implantation was over, the animal was left in a warm environment to recover from anesthesia.

#### 2.4.4. In Vivo MRI Assessment of Tumor Evolution

The animals were subjected to MRI acquisitions to measure tumor volumes and evolution. Acquisitions were performed with 7T BioSpec 70/30 USR spectrometer (Bruker BioSpin GmbH, Ettlingen, Germany) at Servei de Ressonància Magnètica from UAB (also part of Unit 25 of ICTS NANBIOSIS, NMR: Biomedical Applications I https://www.nanbiosis.es/portfolio/u25-nmr-biomedical-application-i/, accessed on 24 November 2021). Mice were positioned in a dedicated bed, which allowed the delivery of anesthesia (isoflurane, 1.5–2% in O_2_ at 1 L/min), with an integrated heating circuit for body temperature regulation. Respiratory frequency was monitored with a pressure probe and kept between 60–80 breaths/min. GL261 tumor-bearing mice were scanned to acquire high-resolution T2w images using a rapid acquisition with relaxation enhancement (RARE) sequence to detect the brain tumors. To calculate the tumor volume from MRI acquisitions, ParaVision software was used to generate regions of interest (ROIs) to measure the tumor area in each slice.

#### 2.4.5. Antitumor Efficacy In Vivo

The antitumor efficacy was evaluated for orthotopic GL261 GB-bearing mice. All animals included in the study were confirmed to have suitable GB with homogeneous volume using T2w MRI as described previously. Mice were randomly divided into two groups: control and complex **1**. All agents were given at a dose of 1.5 mg Pt/kg via intranasal administration. Tumor growth inhibition rate (IR_V_) was calculated using the Equation (1), as follows [39]:IR_V_ = (1 − TVt/TVc) × 100%(1)
where TVt and TVc are mean tumor volumes of treated and control group, respectively.

### 2.5. Statistical Analysis

Quantitative data were expressed as mean ± SE. The multiple-group comparison was performed using a one-way ANOVA. Specific comparison between groups was carried out with an unpaired two-tailed Student’s *t*-test. Differences were considered statistically significant at *p* < 0.05.

## 3. Results and Discussion

### 3.1. Synthesis and Characterization

The synthesis of complex **1** involves three main steps (see Figure 1): (1) oxidation of the commercially available Pt(II) cisplatin (CDDP) with hydrogen peroxide to yield *c*,*c*,*t*-Pt(NH_3_)_2_Cl_2_(OH)_2_ (oxoplatin), (2) addition of succinic anhydride to oxoplatin, using a protocol already described in the literature [36], to yield *c*,*c*,*t*-Pt(NH_3_)_2_Cl_2_(O_2_CCH_2_CH_2_COOH)_2_ (DSCP) and (3) coupling of dopamine (DA) to DSCP via a typical DCC/NHS coupling using N-methylmorpholine (NMM). Following this synthetic route, complex **1** was obtained and purified with a final overall yield of 24.7%. In spite of several attempts, no single crystals suitable for X-ray diffraction were obtained. However, the high-resolution mass spectrum showed the expected mass of complex **1** (see Appendix A). The FT-IR spectra of initial oxoplatin showed peaks at 559 cm^−1^ and 3516 cm^−1^, characteristic of the Pt-O and O-H vibrations, respectively. These peaks disappeared when carboxylic groups of succinic acid react to generate the DSCP complex [40], while new peaks in the 1630–1734 cm^−1^ and 1180–1347 cm^−1^ regions appeared and were assigned to the C-O stretching from the carboxyl groups and succinic acid, respectively. Finally, the carboxyl peak shifted to lower wavenumber (1527–1645 cm^−1^) for complex **1**, indicating the formation of an amide bond with dopamine while a new peak at 1198 cm^−1^ from the C-O stretching of the catechol also appeared (see Appendix A). ^1^H-NMR chemical shifts unequivocally identified the proton signals for complex **1**; meanwhile, all 12 different carbon signals for the symmetrical molecule were found in the ^13^C NMR spectrum (see Appendix A). ^195^Pt NMR spectrum showed a single peak at 1012 ppm (see Appendix A). The chemical stability of the resulting complex was studied in phosphate-buffered saline (PBS, pH 7.4) at 37 °C and PBS/BSA solution using UV-Vis (see Appendix A), NMR and mass spectroscopy as characterization techniques. The non-variation of the initial spectra denotes a stability in the solution superior to that at 24 h with no sign of product reduction.

### 3.2. In Vitro Studies

#### 3.2.1. Cytotoxicity and Cell Internalization

The cytotoxicity of complex **1** at 24 and 72 h was evaluated using the PrestoBlue assay in the murine GL261 GB cell line as well as in human cervical cancer cell HeLa and non-malignant human fibroblast cell 1Br3G, for comparison purposes. These studies were repeated with CDDP as a gold drug model. Both drugs exhibited concentration-dependent cytotoxic effects (see Figure 2a and Appendix A). The half maximal inhibitory concentrations (IC_50_), referred to Pt concentration, were calculated using GraphPad Prism (version 7.0) and are summarized in Table 1.

At 24 h, complex **1** showed lower IC_50_ values than CDDP in both Hela and GL261 tumor cell lines (a reduced cytotoxicity was found for both complexes in the non-tumoral cell line 1Br3G), as confirmed with internalization studies in the GL261 cell line (see Figure 2b), showing that internalization of CDDP at 24 h was notably more effective than that of complex **1.** This fact was attributed to the more effective cellular uptake of CDDP thanks to the favorable interaction between the negatively charged cellular membrane with the positive charge resulting upon hydrolysis of CDDP (under the same experimental conditions, complex **1** is mostly deprotonated with a negative charge that originates electrostatic repulsions) [41]. Remarkably, toxicities at 72 h were comparable.

It is well known that Pt(IV) prodrugs are kinetically inert, thus minimizing unwanted side interactions with biomolecules prior to DNA binding, making them more effective even at lower concentrations [42,43], but they undergo a slower intracellular reduction to produce the active Pt(II) species, as they exhibit more resistance to ligand substitution reaction than Pt(II) centers. Then, the present results suggest that for the first 24 h the activation of Pt(IV) was not very fast, while at 72 h the activation rate appeared sufficient to obtain a similar cytotoxic effect even though the concentration of **1** was comparatively lower. We cannot discard other internalization effects, though the comparative uptake between CDDP and complex **1** at 72 h could not be obtained due to the low cell viability at this time point, which hampers the obtainment of reliable results. It also turns out to be extremely difficult to properly correlate the in vitro and bloodstream biotransformation of Pt(IV) prodrugs as many biomolecules can influence the reaction [44]. Studies using satraplatin, a Pt(IV) complex similar to complex **1**, showed a short half-life of only 6.3 min in whole blood as a result of rapid reduction of the Pt(II) species, being much more stable in fresh plasma (t_1/2_ = 5.3 h) [45]. These results suggest that the cytotoxicity assays could be a reference on how to correlate the cellular uptake with the cytotoxic effect, but the therapeutic potential should be determined by in vivo tests (vide infra).

Finally, and worthwhile to mention, the negative effect on internalization only accounts upon comparison with positively charged drugs. However, if the toxicity of complex **1** is compared with that of the DSCP precursor differing in the presence of carboxylic groups instead of the catechol ones, the scenario differs. In this case, the catechol moiety may increase the lipophilicity of the Pt(IV) prodrug and introduce partially ionizable groups at physiological conditions. In fact, studies of medicinally relevant catechol-derivative compounds revealed that partitioning between the octanol–water phases does not correlate well with the partitioning between the lipid membrane and water, especially for weakly or moderately lipophilic compounds with a combination of ionized and polar chemical groups [46]. The fact of carrying a net positive or negative charge in physiological pH increases the interaction of catecholic compounds with the charged moieties of the lipid headgroups. Moreover, dynamics of the interaction with cell membranes include complex hydrogen bonding with both lipids and water molecules. In our case, the increase in lipophilic character in addition to the presence of ionizable chemical groups seem to have a notable influence in cellular uptake. Other relevant considerations about the results of Table 1 are that the IC_50_ values of NDGA were smaller than those of DSCP but still far (with almost one order of magnitude higher) from those of complex **1**, pointing out an effective synergism between Pt ions and catechol groups.

#### 3.2.2. Pt Bound to DNA

The quantification of Pt bound to DNA can give a direct hint to the anticancer activity. Thus, the three cell lines were treated with complex **1** and CDDP at a concentration of 10 μM (based on Pt content) for 24 h. The DNA concentration was determined by UV-vis spectrometer (Appendix A), while Pt was quantified by ICP-MS. The analysis suggested that in the case of CDDP, the Pt bound to DNA was significantly higher than that found for complex **1** within all cell lines (Figure 2c), reinforcing the trend observed from cellular internalization studies, and its higher anticancer activity for 24 h. As in the case of uptake studies, Pt-DNA binding studies at 72 h were not conclusive due to the low cell viability, which prevented the obtainment of a precise quantification of Pt bound to DNA. An attempt with a lower Pt concentration (1 μM) also failed as measurements were close to the detection limit, resulting in inconclusive values. Additionally, it is worthwhile to mention the accumulation of Pt in nuclei was remarkably higher in GL261 cells than in the fibroblast 1Br3G cells, in line with the low activity of these compounds against 1Br3G cells. As a normal fibroblast cell line, related proteins, i.e., ATP7A and ATP7B, are expressed more in 1Br3G cells than in glial cells, while the expression level of MRP2 is negligible in both [43]. These transporters have been proven to contribute to the efflux of Pt agents and the emergence of drug resistance [42,47]. Different expression levels of these transporters in specific cell lines might account for the distinct Pt accumulation in different cell lines.

#### 3.2.3. Oxidative Stress Assay

Typically, NDGA and overall catechol ligands generate reactive oxygen species (ROS) as a byproduct during the process of oxidation. In fact, dopamine and NDGA itself have already been reported to exhibit cytotoxic effects at a high concentration in various cell lines mainly related to the generation of ROS [43,47,48]. So, to discard oxidative stress as the origin of the notable toxicity of complex **1**, further ROS assays were performed using a fluorescent probe 2′,7′-dichlorofluorescin diacetate (DCFCDA) in the same three cell lines. For comparison purposes, we also tested CDDP, NDGA and H_2_O_2_ under the same experimental conditions (Figure 3). Interestingly, the fluorescence emission triggered by complex **1** and CDDP in all three cell lines was similar and much lower in comparison with the positive control H_2_O_2_, discarding intracellular generation of ROS. Thus, toxicities of both complexes were attributed to DNA binding to exert cell-killing ability which may be the base of the chemotherapeutic effect. It is also remarkable the high ROS response was generated by NDGA in comparison with complex **1** under the same experimental conditions, proving once again the competence of our system.

### 3.3. In Vivo Studies

#### 3.3.1. Tolerability Assessment in Mice

First, in vivo experiments aimed to study the short-term safety and tolerability of complex **1** after IN administration in healthy C57BL/6J female mice (*n* = 3). Three single-scaled doses, namely 0.9, 1.2 and 1.5 mg Pt/kg body weight, were administered weekly, increasing dosage every week. Even with the highest dosage of 1.5 mg Pt/kg, all mice maintained good health status during the study. The body weights of the mice were tracked three times a week with no significant weight decrease indicative of toxicity (see Appendix A). Moreover, no treatment-related adverse effects on food/water consumption, or other clinical signs of suffering were observed during this period. In addition, no obvious morphological/pathological changes were observed in histopathological analyses.

#### 3.3.2. Efficacy Evaluation In Vivo

The in vivo efficacy of complex **1** was evaluated in immunocompetent orthotopic GL261 glioma-bearing mice. The therapy started at day 10 post-implantation (p.i.) with an average tumor volume of 6.15 ± 0.71 mm^3^ and no significant differences in tumor volume or body weight between the control and treated groups (*n* = 7 and *n* = 3, respectively). It was followed by an every-six-days immune-enhanced metronomic schedule (IMS) (Appendix A), which was based on the mice immune cycle and was demonstrated to improve therapeutic outcomes [49,50], most likely due to spurring the host immune system for tumor fighting. Mice were treated with a dose of 1.5 mg Pt/kg body weight, the highest safe dose assessed by tolerability studies. Evolution of tumor volumes was monitored by high-resolution T2w MRI acquisition (see Appendix A), and the corresponding results are shown in Figure 4. Overall, the results (Figure 4a) suggest that complex **1** exhibited a tumor growth inhibition rate (IR_V_) of 20% in comparison with the control group. The survival averages (Figure 4c) of the control group and complex **1** group were 19.9 ± 0.7 days and 22.7 ± 1.7 days, respectively. Whenever mice body weights dropped under 80% of the initial values, experiments were halted, and the mice were euthanized.

In an attempt to improve the therapeutic efficacy, a second round of therapy was launched. For this, a new cohort of mice bearing GL261 tumors was generated (average tumor volume of 5.80 ± 0.94 mm^3^ at day 10 p.i.), without significant differences in tumor volume or body weight between treated and control groups (*n* = 6 each) at the initial time point. Then, mice were treated every 3 days with the same safe dose, 1.5 mg of Pt/kg (Appendix A) [51,52,53]. In this case, as shown in Figure 4d, tumor growth was significantly inhibited even with a higher IR_V_ of 37% in comparison with control group in a shorter period of time. However, the average survival of 19.8 ± 0.2 days was shorter than that in the previous set of experiments and similar to the control group (of 19.9 ± 0.7 days) (Figure 4f), as a total body weight loss equal or superior to 20% total led to euthanasia. Moreover, as can be also seen in Figure 4, the time elapsed for doubling tumor volume (i.e., “doubling time”) decreased for both treatment schedules in comparison with controls (9–11% decrease for the time frame measured, 10 to 16/17 days p.i.). All these results evidenced the effectiveness of IN treatment of GB with complex **1**.

Noteworthily, the application of cisplatin for GB treatment is considerably limited by its poor solubility in aqueous solution, which is ideally 2.53 mg/mL in water at room temperature, but much lower in saline (the concentration of clinical formulation of cisplatin is only 1 mg/mL in 0.9% saline comprising mannitol) [54]. Given the low volume applied for intranasal administration (up to 12 μL), the required concentration of cisplatin to achieve therapeutic effects could not be reached. In contrast, higher concentrations can be achieved using the Pt(IV) prodrug, complex **1,** with improved solubility in saline (around 15 mg/mL, 5.6-fold of CDDP). Besides, increased mucopenetration, due to the presence of catechol, can enhance the accumulation of complex **1** in brain and thus in GB tumors, therefore improving the anticancer efficacy and minimizing the side effects. In addition to the well-known nephrotoxicity, neurotoxicity and hepatotoxicity induced by Pt-based anticancer drugs, damage in fast-growing tissues such as mucous membranes has been widely studied and reported [55]. Therefore, the restricted intranasal delivery of Pt-based drugs can be anticipated in the treatment of GB. To date, few studies have reported to evaluate the in vivo performance of Pt-based agents against GB, although their significance has been recognized to repurpose or reposition the existing chemotherapeutic drugs for GB treatment [56,57]. In the sparse reports, only two cases administered the Pt agents via IN route, all about nanoformulations based on either carboplatin or FePt alloy nanoparticles [58,59]. However, while no in vivo efficacy was involved in either case, muco- or nasal-penetration were. To our knowledge, this work is the first report evaluating Pt agents’ in vivo therapeutic efficacy via IN administration, which shall shed some light on further studies.

## 4. Conclusions

We have reported the synthesis and characterization of complex **1**, which exhibits comparable cytotoxicities to CDDP in tumor cells, specifically in the GL261 GB cell line, while being less toxic against normal fibroblast cells. Even more relevant was the response to its treatment via the IN route in an immune-competent aggressive preclinical (murine) GB model using two different administration protocols, namely an IMS (every-six-days) or in an intensive every-three-days treatment schedule. Significant inhibition on GB tumor growth was observed in both cases, although it was more evident for the intensive schedule. However, the intensive schedule did not improve overall survival time, probably due to an enhanced toxicity in comparison with IMS schedule. On top of that, the axial catecholic ligands endowed the Pt(IV) prodrug with augmented solubility anfd stability in an aqueous solution, which can help overcome one of the formidable limitations for the application of Pt agents for GB treatment. This may be especially relevant in cases such as IN administration, in which therapy is administered in low volumes. Altogether, our findings confirm the potential of nose-to-brain delivery for GB treatment and repurposing Pt-based drugs as GB chemotherapeutic agents. To our knowledge, this is the first report evaluating the in vivo therapeutic efficacy of Pt-based drugs via IN administration, which shall shed some light on further studies

## Data Availability

Data are available on the request from the corresponding author.

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
