# Peer review of "Synthesis and Validation of a Bioinspired Catechol-Functionalized Pt(IV) Prodrug for Preclinical Intranasal Glioblastoma Treatment"

_cancers, 2022, doi:10.3390/cancers14020410_

Round 1
Reviewer 1 Report
In this paper, the synthesis and characterization of a Pt(IV) compound based on cisplatin scaffold and catechol kind ligands in axial positions. The complex was evaluated as anticancer agent with interesting results for preclinical intranasal Glio-3 blastoma treatment. Despite the very nice biological applications that overtake some cisplatin drawbacks, the characterisation of the drug and in particular the stability of this compound in the biological conditions is not sufficiently reported (are not reported at all). The following aspects are missing and must be evaluated before any biological evaluation:
1) Purity of the compound: the authors should perform Elemental Analyses and HPLC (Mass and NMR are not enough indeed). Also 195Pt NMR should be reported;
2) Verify the proof of concept. As shown in Figure 1, the complex analysed should be intra-cellular reduced, releasing cisplatin and the catechol ligands. Did the author verify that this mechanism is happening? Are the catechol lgands released? How fast? Where exactly? It is well accepted that Pt(IV) species are considered prodrugs and are reduced inside the cell but this should be proved for this system and the rate of the reduction analysed and correlated with the biological timing essays (HPLC for instance).
3) Stability in physiological conditions. Even if the authors state in the conclusion (row 470) write that the catechol ligands improved the water solubility, this aspect is not shown and is not studied.
Many other aspects should be also considered or at least discussed. It is well known that ideal Pt(IV) compounds with anticancer activity should have only one modified axial lgand and an OH on the other axial position. This is important in term of reduction kinetics (point 2). Two modified axial ligands sometimes confer a very high stability and the axial ligands are not released in biological timing (24 or 48 h). The study of an analogous compound with only one cathecol ligand would be important and would add value to this work.
Drug uptake. Surprisingly, the cellular uptake of this compound is lower compared to cisplatin. Pt(IV) compounds are more lipophilic than cisplatin in general and they are internalised with a higher extent. The authors should comment on this and determine the lipophilicity index (log P).
To summarize, even if the biological studies look very nice and promising, if the nature, the stability and the purity of the compound is not sufficient, all the studies loose significance.
Reviewer 2 Report
The paper reports the synthesis of a Pt(IV) complex suitable for intranasal administration. The aim was to use such a complex for the treatment of Glioblastoma (one of the most malignant type of brain tumors in adults) for which the standard platinum-based drugs (and also other drugs) are ineffective because of their difficulty to cross the blood-brain barrier and achieve effective therapeutic concentration at the tumor sites. Intranasal administration could overcome the problem of delivery into the brain. Catechol functionalization had already been exploited to promote mucopenetration, thus a Pt(IV) prodrug functionalized with catecholic moieties (complex 1) was synthesized and its biological activity investigated. Complex 1 showed a cytotoxic effect comparable to that of cisplatin in HeLa, 1Br3G and GL261 cells. Furthermore, preclinical in vivo therapy using intranasal administration route suggested that it can reach the brain and inhibit the growth of orthotopic GL261 glioblastoma.
The synthesis of the new complex was straightforward and its characterization is satisfactory.
In HeLa and GL261 cells, the cytotoxic effect of compound 1 was lower than that of cisplatin at 24 h incubation time, but comparable at 72 h incubation time. Consistently, the Cellular uptake and the DNA-bound Pt were lower for Compound 1 than for cisplatin at 24 h incubation time. Unfortunately, cellular uptake and DNA-bound Pt were not measured at 72 h incubation time because of low cell viability at that time point. However I don’t see why the authors have not attempted an experiment at 72 h incubation time using a lower concentration of drug.
The in vivo efficacy of complex 1 was evaluated in immunocompetent orthotopic GL261 glioma-bearing mice using two different administration protocols: an every-6-days immune enhanced metronomic schedule and an intensive every-3-days treatment schedule. Significant inhibition on GB tumor growth was observed in both cases, more significant for the intensive schedule. However, the intensive schedule did not improve the overall survival time, probably due to an enhanced toxicity in comparison with the immune enhanced metronomic schedule. In my opinion, in order to confirm the real potential of nose-to-brain delivery of compound 1 for GB treatment, the authors should perform also an experiment in which compound 1 is administered to the mice by the classical intravenous injection and compare the results with those of the intranasal administration.
The paper is well written and easy to read. However in a few occasion there is a strange symbol before M or L (is it a symbol standing for micro?).
Concerning the activation of Pt(IV) prodrugs in a physiological setting I would suggest to cite a very recent and comprehensive investigation recently appeared in Angewandte
https://doi.org/10.1002/ange.202114250
Round 2
Reviewer 1 Report
The authors, addressed most of the previous concerns and/*or provided a a very clear and rationale explanation when they didn't.
I think the paper can be accepted for publication. Please, add the 195Pt NMR spectrum in the SI
Reviewer 2 Report
The authors have tried to address the first of the two major points raised in the first review, however, because of technical limitations, the question couldn’t be answered.
Concerning the second point raised in the first review, the authors reply “there is already a bunch of published literature supporting dose-limiting toxicities and ineffective delivery as the main problems related to platinum compounds to treat brain tumors via intravenous injection. I don’t find the answer satisfactory since compound 1 is a Pt(IV) complex and complexes of this type are much more stable than Pt(II) counterparts and also much more lipophilic. I understand that to run an in vivo test is cumbersome and expensive, however I would strongly suggest the authors to run such an experiment in the future investigation.